# Predictors of Progression in a Series of 81 Adult Patients Surgically Managed for an Intracranial Hemangioblastoma: Implications for the Postoperative Follow-Up

**DOI:** 10.3390/cancers16071261

**Published:** 2024-03-23

**Authors:** Elisabeth Garrido, Huy Le Ngoc, Jacques Guyotat, Isabelle Pelissou-Guyotat, Timothée Jacquesson, Violaine Delabar, Romain Manet, Clémentine Gallet, Tanguy Fenouil, Nathalie Streichenberger, Alexandre Vasiljevic, David Meyronet, Emmanuel Jouanneau, François Ducray, Chloe Dumot, Thiebaud Picart

**Affiliations:** 1Department of Neurosurgery, Rouen University Hospital, 1 Rue de Germont, 76000 Rouen, France; elisabeth.garrido@chu-rouen.fr; 2Department of Neurosurgery, Hospital Bach Mai, 78 Giai Phong, Phuong Mai, Dong Da, Ha Noi 116305, Vietnam; lnh@bachmai.edu.vn; 3Department of Neurosurgery, Hôpital Neurologique Pierre Wertheimer, Groupe Hospitalier Est, Hospices Civils de Lyon, 59 Boulevard Pinel, 69500 Bron, France; jacques.guyotat@chu-lyon.fr (J.G.); isabelle.pelissou-guyotat@chu-lyon.fr (I.P.-G.); timothee.jacquesson@chu-lyon.fr (T.J.); violaine.delabar@chu-lyon.fr (V.D.); romain.manet@chu-lyon.fr (R.M.); clementine.gallet@chu-lyon.fr (C.G.); emmanuel.jouanneau@chu-lyon.fr (E.J.); chloe.dumot@chu-lyon.fr (C.D.); 4Faculty of Medicine Lyon Est, Université Claude Bernard Lyon 1, 8 Avenue Rockefeller, 69003 Lyon, France; tanguy.fenouil@chu-lyon.fr (T.F.); nathalie.streichenberger@chu-lyon.fr (N.S.); alexandre.vasiljevic@chu-lyon.fr (A.V.); david.meyronet@chu-lyon.fr (D.M.); francois.ducray@chu-lyon.fr (F.D.); 5Cancer Research Centre of Lyon (CRCL) INSERM 1052, CNRS 5286, 28 Rue Laennec, 69008 Lyon, France; 6Department of Neuropathology, Groupe Hospitalier Est, Hospices Civils de Lyon, 59 Boulevard Pinel, 69500 Bron, France; 7CNRS UMR 5310—INSERM U1217, Institut NeuroMyogène, 8 Avenue Rockefeller, 69008 Lyon, France; 8Department of Neuro-Oncology, Hôpital Neurologique Pierre Wertheimer, Groupe Hospitalier Est, Hospices Civils de Lyon, 59 Boulevard Pinel, 69500 Bron, France; 9CarMeN Laboratoire, INSERM, INRAER, Université Claude Bernard Lyon 1, 59 Boulevard Pinel, 69500 Bron, France

**Keywords:** extent of resection, hematocrit, hemangioblastoma, progression-free survival, von Hippel–Lindau disease

## Abstract

**Simple Summary:**

Local or distant progression is possible after the resection of an intracranial hemangioblastoma. Few studies have focused on predictors of hemangioblastoma progression, especially for intracranial locations. Therefore, the aim of the present study was to precisely identify the predictors of local and distant progression in a series of 81 patients managed for an intracranial hemangioblastoma in order to ultimately tailor the follow-up to each patient profile. As about a quarter of the patients with sporadic hemangioblastoma would have a local progression, with a progression-free survival of 56 months following surgery, it would be advisable to plan a regular surgical and radiological screening for at least 10 years postoperatively. The local recurrence can be particularly quick in the case of partial resection, which justifies closer radiological monitoring. In patients with von Hippel–Lindau disease, annual monitoring has to be planned indefinitely given the risk of local and distant progression.

**Abstract:**

The aim was to identify predictors of progression in a series of patients managed for an intracranial hemangioblastoma, in order to guide the postoperative follow-up modalities. The characteristics of 81 patients managed for an intracranial hemangioblastoma between January 2000 and October 2022 were retrospectively analyzed. The mean age at diagnosis was of 48 ± 16 years. Eleven (14%) patients had von Hippel–Lindau disease. The most frequent tumor location was the cerebellar hemispheres (*n* = 51, 65%) and 11 (14%) patients had multicentric hemangioblastomas. A gross total resection was achieved in 75 (93%) patients. Eighteen (22%) patients had a local progression, with a median progression-free survival of 56 months 95% CI [1;240]. Eleven (14%) patients had a distant progression (new hemangioblastoma and/or growth of an already known hemangioblastoma). Local progression was more frequent in younger patients (39 ± 14 years vs. 51 ± 16 years; *p* = 0.005), and those with von Hippel–Lindau disease (*n* = 8, 44% vs. *n* = 3, 5%, *p* < 0.0001), multiple cerebral locations (*n* = 3, 17% vs. *n* = 2, 3%, *p* = 0.02), and partial tumoral resection (*n* = 4, 18% vs. *n* = 1, 2%, *p* = 0.0006). Therefore, it is advisable to propose a postoperative follow-up for at least 10 years, and longer if at least one predictor of progression is present.

## 1. Introduction

Hemangioblastomas are rare and benign tumors, classified as vascular mesenchymal, non-meningothelial tumors, according to the WHO 2021 classification of tumors of the central nervous system [1]. They represent 1 to 2% of cerebral tumors and 2 to 10% of primary spinal cord tumors [2]. These highly vascularized neoplasms may occur sporadically, or as part of von Hippel–Lindau (VHL) disease in about 20–30% of cases [3]. They mainly affect young people, between 20 and 40 years old [4], and their most frequent location is the posterior fossa [2,5].

Arvid Vilhelm Lindau historically described hemangioblastomas as lesions with an “unmistakable neoplasticity”, composed of “blood vessel elements”, and with a “tendency towards cyst formation” [6]. Although hemangioblastomas do not metastasize, they can be proliferating, multifocal, and cause neurological deficits. Most hemangioblastomas can be totally resected safely. Nevertheless, some of these lesions recur despite a well-conducted treatment [2]. 

It is very important to follow patients postoperatively given not only the risk of local progression, but also of the appearance of de novo hemangioblastomas located at a distant site. Yet, there are no clear recommendations regarding follow-up modalities, and medical practices may vary depending on the center and the neurosurgeon’s habits. Few studies have focused on predictors of hemangioblastoma progression due to the rarity of these tumors [7]. The aim of the present study was to identify predictors of local and distant progression in a series of patients managed for an intracranial hemangioblastoma in order to ultimately tailor the follow-up to each patient profile. 

## 2. Materials and Methods

### 2.1. Data Collection

The Hospices Civils de Lyon database was screened to retrospectively identify consecutive patients surgically managed for an intracranial hemangioblastoma between 1 January 2000 and 1 October 2022. All cases of hemangioblastoma were histologically confirmed by a senior neuropathologist. Pediatric cases (<18 years), isolated spinal hemangioblastomas, and patients with less than a one-year follow-up were excluded. Medical records, available imaging, and operative reports were reviewed for each patient. 

Demographics data, radiological features (tumor location, tumor size, hydrocephalus, presence of multiple hemangioblastomas), treatment modalities (pre-operative embolization, total or partial tumoral removal, radiotherapy), histological features, pre- and early postoperative (<72 h) hematocrit levels, date and mode of progression, if any, and date of last radiological follow-up were collected. All patients in whom VHL disease was suspected based on clinical observation underwent a genetic analysis. Only patients harboring a VHL germline mutation were considered true cases of VHL disease (*n* = 11). For patients who received preoperative embolization and radiotherapy, the indication was validated during a tumor board meeting. 

### 2.2. Imaging Features

On the basis of magnetic resonance imaging (MRI), tumoral location was classified into five categories: cerebellar hemisphere, cerebellar vermis, floor of the fourth ventricle, ponto-cerebellar angle, and supra-tentorial. The classification of Resche was used to describe the radiological presentation of hemangioblastomas that were classified into four types [8]. Type 1 hemangioblastomas (simple cyst type) are characterized by a cyst with clear fluid and smooth walls, without evidence of a mural nodule on angiographic sequences. Type 2 hemangioblastomas (macrocystic type) are the most frequent and are characterized by a cyst of variable size with a mural nodule. Type 3 hemangioblastomas (solid type) have a solid consistency with blurred limits and marked vascularization. Type 4 hemangioblastomas (microcystic type) are solid but contain small cysts measuring a few millimeters. The extent of tumor resection was assessed on postoperative MRI, and total resection was defined as the absence of residual contrast enhancement on the gadolinium contrast-enhanced T1-weighted sequence according to the radiologist’s analysis.

### 2.3. Histological Features

Histologically, hemangioblastomas are composed of four cell types: endothelial, stromal and mast cells, and pericytes. Only stromal cells are tumoral cells and their cytoplasm can be vacuolated (lipid-laden cytoplasm) [6]. Moreover, stromal cells can be arranged in two variants: a reticular type, with a cellular distribution around the vascular network, and a cellular type, with cells grouped in larger sheets or clusters [6]. Both subtypes are sometimes present in the same sample. The arrangement of the stroma was noted. The presence of lipid-laden stromal cells was recorded, according to the evaluation of the neuropathologist.

### 2.4. Survival

The progression-free survival was defined as the time between tumor removal and the first brain imaging confirming progression or the last known brain imaging if the patient did not recur. For surviving patients, this interval was censored at the date of last imaging follow-up. 

Local progression, defined by tumor regrowth in the resection cavity, was distinguished from distant progression, which corresponded to the appearance of a new hemangioblastoma at a site different from that of the initially resected hemangioblastoma and/or the growth of an unresected hemangioblastoma, already present on the first imaging. For patients with local progression, the Resche type of the recurrent tumor (which could be different from this of the initial tumor), hematocrit level and treatment modalities at progression were collected. For patients with distant progression, tumor location, Resche type, and treatment modalities were consigned but the hematocrit level was available in too few patients to be taken into account in the analysis.

### 2.5. Statistical Analyses

Descriptive data were presented as the mean and standard deviation for quantitative variables, except for progression-free survival, which was expressed as a median with the smallest and largest values. Effectives and percentages were used for qualitative variables. Statistical analyses were conducted using JMP software v14.1.0 (SAS institute). Computer-assisted statistical analysis was performed with the two-tailed χ^2^ test or the Fisher exact test, as appropriate, for categorical variables, and the Mann–Whitney U test for continuous variables. The actuarial data were represented with Kaplan–Meier plots and comparison was performed with the log-rank test. The statistical tests were bilateral and a *p*-value less than 0.05 was considered statistically significant.

### 2.6. Standard Protocol Approvals and Registrations

Study design and manuscript organization were guided by the STROBE statement on cohort studies. This study was conducted in accordance with the local and international ethical standards, as well as the 1964 Helsinki declaration and its later amendments. All patients provided informed consent for tumor sample inclusion in the Hospices Civils de Lyon biological resource center and gave informed consent for the retrospective extraction of their clinical data.

## 3. Results

### 3.1. Patient Characteristics 

A total of 81 patients were included. The mean age at diagnosis was of 48 ± 16 years (range, 18–81 years) (Table 1). The sex ratio male/female was 1.02 and most patients were Caucasians (94%). Eleven (14%) patients had genetically confirmed VHL disease. 

Radiologically, the most frequent tumor location was the cerebellar hemispheres (*n* = 51, 65%), followed by the floor of the fourth ventricle (*n* = 11, 14%), the cerebellar vermis (*n* = 9, 11%) and the ponto-cerebellar angles (*n* = 4, 5%). Only four (5%) hemangioblastomas were developed in the supra-tentorial region. The predominant presentation on the T1 gadolinium-enhanced sequence corresponded to Resche type 2 (macroscystic type, *n* = 54, 72%). The mean tumor size, defined as the largest diameter of the lesion (cyst included) was 37 ± 11 mm (range: 12 to 70 mm), whereas the solid part of the tumor measured, on average, 14 ± 6 mm (range: 7 to 26 mm). Obstructive hydrocephalus was present in 18 patients (27%). In 11 (14%) patients, including 10 (12%) with a genetically confirmed VHL disease, the preoperative MRI scan found one or several associated hemangioblastomas for which immediate surgical resection was not required. Among them, five (6%) patients had multiple intracranial hemangioblastomas. Other hemangioblastomas were spinal (*n* = 9, 11%) and retinal (*n* = 6, 7%). 

Regarding the management, only one (1%) patient had a preoperative embolization given the high tumor volume. A gross total resection (GTR) was achieved in 75 (94%) patients and five (6%) patients had a partial resection. One (1%) patient had an adjuvant radiotherapy because the resection was very partial. 

Concerning the histological features, the stroma had a reticular aspect in 58 (78%) tumors and a cellular aspect in 20 (27%) tumors. Lipid-laden stromal cells were observed in 39 (52%) cases. 

Biologically, the preoperative and the postoperative hematocrit levels were of 42 ± 5% and 39 ± 4%, respectively.

In terms of outcomes, the median progression-free survival reached 132 months 95% CI [108;154], for the whole series (Figure 1). 

A local progression was diagnosed in 18 (22%) patients, with a median progression-free survival of 56 months 95% CI [1;240]. The radiological presentation was modified in two (14%) patients whose tumors, initially classified as Resche types 1 and 4, switched to Resche type 3 at progression (Table 2). The management of the local progression consisted in surgery (*n* = 13, 76%) associated to adjuvant radiotherapy in two patients (*n* = 2, 12%). 

Furthermore, 11 (14%) patients presented a distant progression, which corresponded to new hemangioblastomas (*n* = 7, 64%), growth of an already known hemangioblastoma (*n* = 2, 18%), or both (*n* = 2, 18%), after a median follow-up of 90 months (range: 24 to 240 months). Among these patients with distant progression, seven (9%) also had a local progression (Figure 2). All distant progressive hemangioblastomas were located in the cerebellar hemispheres (*n* = 11, 100%) and their radiological presentation corresponded to Resche type 2 (*n* = 2, 20%) and Resche type 3 (*n* = 8, 80%) (Table 2). Thus, only 22% of these hemanglioblastomas had the same presentation as the removed hemangioblastomas, according to the Resche classification. The management of patients who had a distant progression consisted of surgery (*n* = 5, 45%) or radiotherapy (*n* = 1, 9%) for patients ineligible for tumor resection. The remaining patients were followed radiologically given the small size of the tumor and the absence of symptoms.

### 3.2. Characteristics of Patients with VHL Disease

In order to identify the characteristics of patients with VHL disease, this subgroup was compared to patients without VHL disease (Table 3). Patients with VHL disease were younger (36 ± 14 vs. 50 ± 15, *p* = 0.0045), had more frequent multiple hemangioblastomas at diagnosis (*p* < 0.0001 for each location), local progression (73% vs. 14%, *p* < 0.0001), and distant progression (73% vs. 4%, *p* < 0.0001) than other patients. There was no significant difference regarding the other characteristics. Notably, progression-free survival did not significantly differ in patients with and without VHL disease (132 months 95% CI [104;163] vs. 120 months 95% CI [88;148], *p* = 0.237) (Figure 3a).

### 3.3. Characteristics of Patients with Local Progression 

In order to better define the characteristics associated with local progression, patients who had a local progression were compared to patients who did not recur (Table 1). Concerning the demographic characteristics, patients with local progression were significantly younger (39 ± 14 years vs. 51 ± 16 years; *p* = 0.005) and more frequently had VHL disease than the other patients (*n* = 8, 44% vs. *n* = 3, 5%, *p* < 0.0001). There was no difference regarding gender (*p* = 0.64) and ethnicity (*p* = 0.90). 

There was no significant difference in the two groups regarding tumor location, tumor size, and Resche type. Nevertheless, compared to other patients, patients with local progression more frequently had multiple cerebral locations (*n* = 3, 17% vs. *n* = 2, 3%, *p* = 0.04) on their initial MRI scans but there was no significant difference concerning spinal (*n* = 1, 6% vs. *n* = 8, 12%, *p* = 0.12) or retinal locations (*n* = 4, 22% vs. *n* = 2, 3%, *p* = 0.07). 

GTR was performed in almost all patients without progression (*n* = 62, 98%) but in only 72% (*n* = 13) of patients in the progression group. Therefore, a partial tumoral removal was not only a strong predictor of progression (*p* = 0.0006) but was also associated with a dramatic decrease in the median progression-free survival (PFS of 9 months 95% CI [26;44] vs. 132 months 95% CI [106;154], *p* < 0.0001) (Figure 3b). Given the inequal repartition of partial resection between the considered age groups (*n* = 1 under 40 years; *n* = 5 over 40 years), the impact of age on progression-free survival was compared in patients aged over and under 40 years. There was no significant difference (132 months 95% CI [94;164] vs. 108 months 95% CI [78;134], *p* = 0.175) (Figure 3c). Concerning the other therapeutic modalities, no significant difference was found for radiotherapy (*n* = 0, 0% vs. *n* = 1, 2%, *p* = 0.60) or preoperative embolization (*n* = 0, 0% vs. *n* = 1, 2%, *p* = 0.60). Densities of lipid-laden stromal cells had a similar repartition in both groups (*p* = 0.91) but the tumor stroma less frequently had a reticular aspect in patients with progression than in other patients (*n* = 9, 64% versus *n* = 49, 82%, *p* = 0.04). The pre- and postoperative hematocrit levels of the two groups (40 ± 5% vs. 42 ± 5%, *p* = 0.94 and 29 ± 5% vs. 38 ± 7%, *p* = 0.12, respectively) did not significantly differ. However, in patients without progression, there was a significant drop in the hematocrit level from 40 ± 5% preoperatively to 29 ± 5% postoperatively (*p* = 0.003), whereas the difference was not significant for patients with a local progression (from 42 ± 5% to 38 ± 7%, *p* = 0.22). Finally, distant progression was more common in patients with local progression (*n* = 7, 58%) than in other patients (*n* = 11, 16%; *p* = 0.0003). 

### 3.4. Characteristics of Patients with Distant Progression 

In order to identify the characteristics associated with distant progression, patients who had distant progression were compared to patients who did not have distant progression (Table 4). 

Compared to other patients, patients who had a distant progression were younger (37 ± 17 vs. 50 ± 15, *p* = 0.01) and more frequently had VHL disease (*n* = 8, 67% vs. *n* = 3, 4%, *p* < 0.0001). As expected, in the subgroup of patients with distant progression, multiple brain hemangioblastomas (*p* < 0.001), spinal hemangioblastomas (*p* < 0.0001), and retinal hemangioblastomas (*p* < 0.0001) were more frequently found at diagnosis than in the other patients. The histological features of the removed hemangioblastoma did not differ significantly in the two groups. Consistent with previous results, local progression was more frequently diagnosed in patients with distant progression compared to other patients (*n* = 7, 64% vs. *n* = 11, 15%; *p* = 0.002). The follow-up was almost three times longer for patients who had a distant progression than for other patients (90 months vs. 36 months; *p* = 0.02). 

## 4. Discussion

In the present series of 81 patients managed for an intracranial hemangioblastoma, about a quarter of them had a local progression, with a median progression-free survival of 56 months. For patients with VHL disease, the rate of local progression was higher and concerned almost three quarters of the patients (8/11 patients). These data are consistent with previously published series in which the progression rates ranged from 5% to 25% in sporadic cases [2,9,10,11] and from 17% to 75% in patients with VHL disease [11].

### 4.1. Predictors of Local and Distant Progression

Young age, VHL disease, presence of multiple intracranial hemangioblastomas on the first imaging, partial resection, and a reticular arrangement of the tumor stroma were identified as predictors of local progression in the present series. In accordance with previous studies [12,13,14,15], patients with VHL disease were younger at diagnosis and more frequently developed multiple hemangioblastomas, compared to other patients. Despite the absence of multivariate analysis given the too-little progression rates, these data suggest that age and multiple locations are certainly confounding factors associated with VHL disease in the present study.

Although the surgical outcomes have been specifically analyzed for spinal or brainstem locations [7,16,17,18], relatively few recent studies have focused on intracranial hemangioblastomas. In these retrospective series, young age, VHL disease, and the presence of multiple hemangioblastomas in the central nervous system were consistently identified as predictors of recurrence [2,13]. In a recent meta-analysis including 237 supratentorial cases from 169 studies, GTR and cystic radiological presentation were associated with a longer progression-free survival [19]. This trend was not confirmed in the present series, probably because cystic presentation was far more common than solid presentation. A study based on the US National Cancer Database including more than 1000 patients, but not restricted to intracranial locations, identified GTR as a predictor of longer overall survival, but only in patients younger than 40 years [20], which consistently highlights the predictive value of age. Another recent study, based on the Surveillance, Epidemiology, and End Results Database, including extracranial locations, identified multiple tumors as a negative predictor of overall survival [5]. Additionally, age over 60 years, but especially over 80 years, was also associated with a worse prognosis in terms of overall survival [5,20]. This result may consequently not only be linked to an increase in solid forms and tumor size but also to the surgical challenges specific to this subpopulation [11,21]. Too few elderly patients were included in the present series to confirm this. Finally, regarding histological parameters, the characteristic of the tumor stroma has not been previously described as a predictor. However, it was shown that recurrent tumors tended to be enriched in lipid-laden stromal cells [2]. The present results did not sustain this hypothesis.

Young age, VHL disease, and multiple intracranial hemangioblastomas were not only predictors of local but also of distant progression. Extracranial locations and local progression were the other predictors of distant progression. The rarity of multiple locations out of the setting of VHL disease emphasizes the major role played by genetic predisposition in distant progression. To the best of our knowledge, no previous study has been dedicated to the assessment of distant progression in patients surgically managed for hemangioblastoma. However, it was demonstrated that VHL disease patients younger than 20 years had a significantly higher risk of developing a new hemangioblastoma than patients older than 40 years [12,22]. This seems to confirm the predictive value of age, in conjunction with genetic factors.

### 4.2. Genetic Predisposition of Hemangioblastoma Progression

VHL disease is caused by a germline mutation of the VHL gene (located on the short arm of chromosome 3) that encodes the tumor suppressor protein VHL. It is an autonomic dominant neoplasia syndrome, causing visceral and nervous central system lesions [12]. Cranio-spinal hemangioblastomas are present in 60–80% of cases and are responsible for significant morbidity and mortality [12,23]. In these patients, it is also important to note that local recurrent lesions may correspond to new sporadic lesions rather than true recurrences, especially in the case of complete resection.

Of all nervous central system hemangioblastomas, 10–40% are associated with VHL disease [3,5,14,24]. This rate was smaller in the present study, with a confirmed VHL mutation in only 14% of patients, notably because of the exclusion of pediatric and spinal cases. Additionally, it is highly possible that the three patients that had a distant progression (Figure 2) also had VHL disease, where genetic confirmation could not be provided, probably because of the difficulty in diagnosing this disease, especially in cases of mosaicism [25].

In patients with VHL disease, the European and North American guidelines [26,27] recommend performing annual monitoring indefinitely, in accordance with the risk of local but also distant progression, which can occur later than the former, as illustrated in the present series.

### 4.3. Indications Provided by Haematocrit Level Analysis

The significant association between hemangioblastoma and polycythemia is well documented [28,29]. The alteration of the VHL tumor suppressor gene results in the stabilization of hypoxia-induced factor 1/2α, which then increases the biosynthesis of erythropoietin (EPO), and thus leads to polycythemia [30]. In addition to germline mutations in patients with VHL disease, somatic mutations of the VHL gene are also frequently found in sporadic hemangioblastomas [31]. Moreover, EPO production has been identified in the stromal cells of hemangioblastomas, which could explain that the comparison of patients with and without tumor recurrence highlighted no significant difference regarding the pre- and postoperative hematocrit levels, in accordance with previous results [2].

Interestingly, a significant postoperative decrease in the hematocrit level was only identified in patients without tumor progression (29 ± 5% postoperatively vs. 40 ± 5% preoperatively; *p* = 0.003). The most probable explanation is that the intraoperative bleeding was more important in this subgroup, suggesting a more aggressive resection. This is consistent with the higher rate of GTR. Furthermore, it could be hypothesized that the hematocrit level, conditioned by EPO production, could be a biomarker of stromal cells’ presence and activity. Consequently, it could be hypothesized that a drop in hematocrit could be an indicator of the arrest of EPO tumor production and thus of surgical success. However, a later analysis of the postoperative hematocrit level is mandatory to confirm these elements.

### 4.4. Recommendations Regarding Patient Management

From a surgical viewpoint, the rate of GTR was high (93%), and in line with other series of intracranial hemangioblastomas [10,11,32,33]. A partial resection was expectedly a strong predictor of local progression, which was early in most cases (60% in the two years following surgery—Figure 3b). In the present study, the rate of local progression after a GTR was 17% (13/75) and slightly higher than in previous series, in which it ranges from 1% to 10% [6,12,34]. A slight overestimation of the rate of GTR cannot be ruled out, given the relatively bad resolution of the oldest MRI scans. Nevertheless, the progression-free survival of 56 months is comparable to that previously reported [6].

Consequently, for symptomatic hemangioblastomas (sporadic or in the context of VHL disease), it may be advised to perform a safe maximal surgical resection, which remain associated with a better overall survival compared to other treatment modalities [5,15,20,33,34]. In the case of neurological worsening following surgery, functional recovery is more limited in older patients, obese patients, and in solid forms, justifying greater precautions in such cases [11]. Although the cyst walls, if present, were classically considered non-tumoral [35], more recent data seem to indicate otherwise, warranting the recommendation to resect these structures if they exhibit intraoperative fluorescence after 5-ALA intake [36]. Further results are required to be conclusive. Regarding the surgical technique, circumferential dissection with devascularization and en bloc removal is recommended to limit hemorrhagic complications [4]. Preoperative embolization was performed in only one case in the present study. Indeed, this technique does not decrease the rate of intraoperative hemorrhagic complications and can lead to significant complications, including post-procedural bleeding. It is consequently not considered a standard of care for intracranial hemangioblastomas [37].

The place of radiotherapy in the management of intracranial hemangioblastomas is still debated and few results are available in the literature. Historically, radiotherapy was considered an option, especially after partial resection, at progression or for surgically inaccessible lesions [12], as in the present study. According to a survival analysis, patients receiving surgery for the management of a cranial or spinal hemangioblastoma had better overall survival compared with patients receiving both surgery and radiotherapy [5]. Consequently, postoperative adjuvant radiotherapy should not be recommended systematically and only has to be discussed in patients with very partial resection.

Compared to surgery, first-line radiotherapy indeed results in a worse overall survival [5]. Efficacy seems limited in cystic lesions and decreases over time with a 15-year progression-free survival rate of only 51% [12]. The investigations regarding the place of radiosurgery concluded that this technique may also represent an alternative for patients with limited surgical options, especially in the case of VHL disease [38,39,40], with a low rate of local failure (4%), a 5 year progression-free survival rate of 79–92%, and a limited rate of adverse events (7%) [40,41]. The best results in terms of progression-free survival are obtained in patients with VHL disease, a solid tumor, a small tumor volume, and without history of surgical resection [39,41]. A systematic review consistently concluded that gamma-knife could represent an interesting option in patients with VHL disease, in the case of limited surgical alternatives, leading to a tumor control in more than 90% of cases at 5 years. However, the long-term outcomes and the underlying factors associated with tumor progression remain to be assessed [38].

For unresectable tumors, the last therapeutic option could be represented by selective inhibitors of hypoxia-induced factor 2α, which are now approved by the Food and Drug Administration in the United States [42,43,44], or the tyrosine kinase inhibitor pazopanib [45], which, providing promising preliminary results, are still under evaluation in patients with VHL disease. HIF-2α inhibitors are associated with a 30% response rate in hemangioblastomas and a reduction in their size from 20% to 50% over the course of treatment. Anemia is the most common adverse effect. Although long-term results are not known, these treatments represent an interesting option [42,46,47].

For asymptomatic locations, particularly in patients with VHL disease, tumor growth is unpredictable [22,48]. Therefore, given the surgical risks, it is recommended to propose a treatment only in the case of clinical progression [22,48], as intervention solely based on radiological progression would result in up to four additional procedures per patient in a 10-year period [48].

### 4.5. Limitations of the Study

The main limitation of the present study is inherent to its retrospective design. It was not possible to lead a multivariate analysis as the rates of progression were too low to ensure statistical relevance. However, this is one of the largest monocentric studies of intracranial hemangioblastomas, in which patients were managed the most homogenously way possible, thus limiting confounding factors and providing useful indications to tailor the post-operative follow-up according to patient characteristics.

## 5. Conclusions

In patients with sporadic hemangioblastoma, about a quarter of the patients would have a local progression, with a progression-free survival of 56 months following the surgery. Distant progression is exceptional and concerns patients that have a possible false negative VHL disease test. Therefore, it would be advisable to plan a regular clinical and radiological screening (cranial MRI only) for at least 10 years postoperatively. The local recurrence can be particularly quick in the case of partial resection, which justifies a closer radiological screening. Young age and multicentric location are associated with an increased risk of both local and distant progression, which can even occur a very long time after the first management. These two factors are probably not independent from VHL disease and may warrant the same follow-up modalities as the latter.

In patients with VHL disease, the European and North American guidelines [26,27] recommend performing annual monitoring indefinitely, in accordance with the risk of local but also distant progression, which can occur later than the former. This monitoring includes a neurological examination and a cranio-spinal MRI scan every year, in addition to a clinical eye inspection, a hearing exam, and a plasma metanephrin and normetanephrine dosage [26,27].

Finally, the place of hematocrit and EPO level as predictive biomarkers of progression remains to be defined.

## Figures and Tables

**Figure 1 cancers-16-01261-f001:**
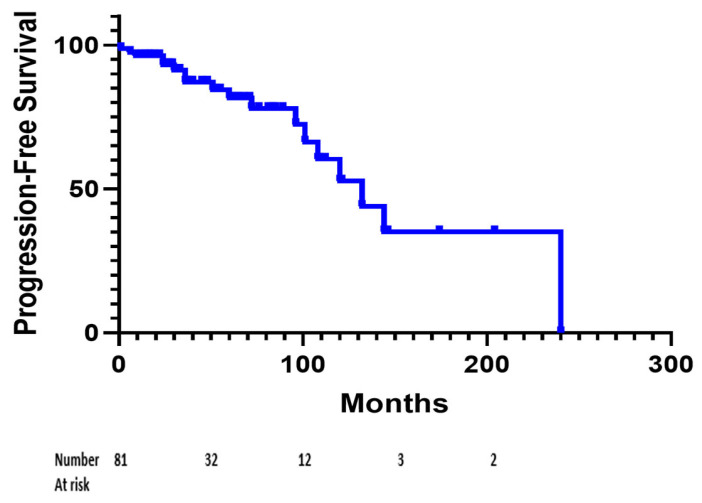
Kaplan–Meier analysis for progression-free survival in the whole series of patients managed for an hemangioblastoma. Sixty-three patients were censored at the date of the last follow-up.

**Figure 2 cancers-16-01261-f002:**
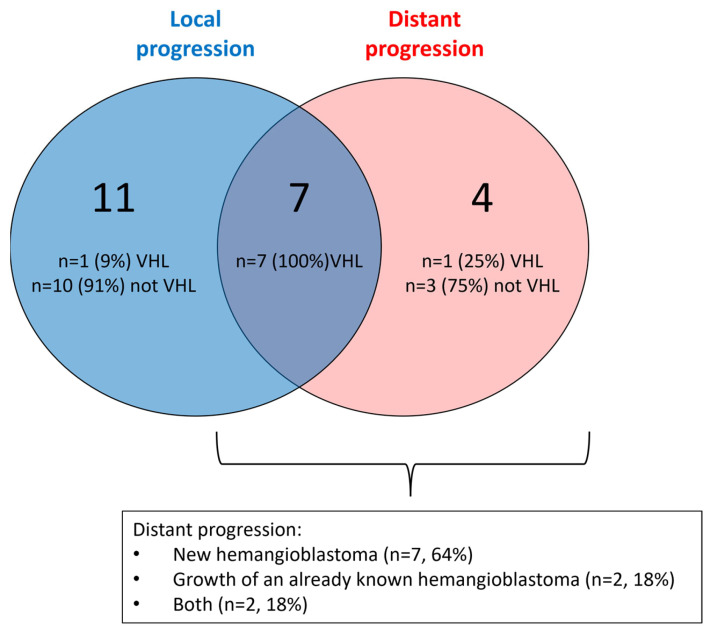
Repartition of the modes of progression.

**Figure 3 cancers-16-01261-f003:**
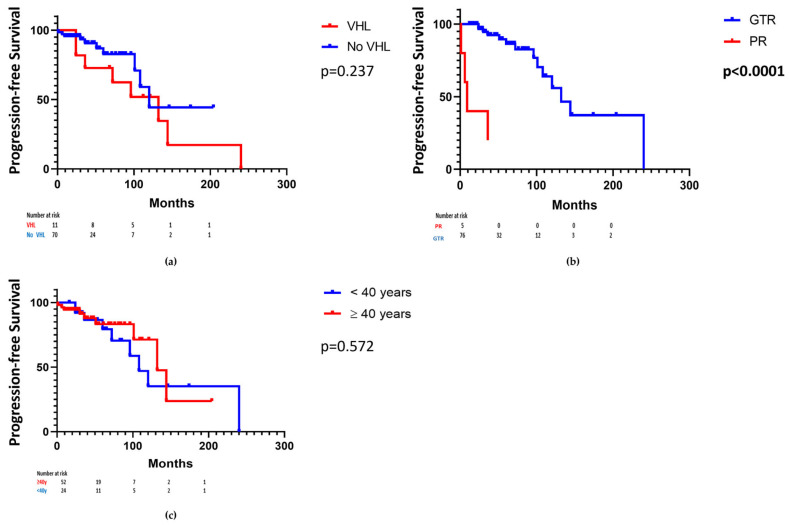
Kaplan–Meier analysis for progression-free survival (**a**) according to von Hippel–Lindau (VHL) status; (**b**) according to the quality of resection; (**c**) according to the age (<40 years or ≥40 years) after the exclusion of patients with partial resection.

**Table 1 cancers-16-01261-t001:** Characteristics of the whole series of patients surgically managed for a hemangioblastoma and comparison of patients with and without local progression.

	Whole Series n = 81	Local Progression n = 18	No Local Progression n = 63	*p*-Value ^3^
**Demographic Data**				
**Age** (years) mean ± SD	48 ± 16	39 ± 14	51 ± 16	**0.005 ^4^**
**Sex**				0.64 ^5^
Male	41 (51)	10 (56)	31 (49)	
Female	40 (49)	8 (44)	32 (51)	
Caucasian				0.90 ^6^
Yes	76 (94)	17 (94)	59 (94)	
No	5 (6)	1 (6)	4 (6)	
**VHL disease**				**<0.0001 ^6^**
Yes	11 (14)	8 (44)	3 (5)	
No	70 (86)	10 (56)	60 (95)	
**Radiological Features**				
**Tumor location**	n = 79	n = 17	n = 62	
Cerebellar hemisphere	51 (65)	9 (53)	42 (68)	0.20 ^5^
Vermis	9 (11)	4 (24)	5 (8)	0.09 ^6^
Floor of the fourth ventricle	11 (14)	1 (6)	10 (16)	0.26 ^6^
Ponto-cerebellar angle	4 (5)	2 (12)	2 (3)	0.17 ^6^
Supra-tentorial	4 (5)	1 (6)	3 (5)	0.89 ^6^
**Resche Classification**	n = 75	n = 15	n = 60	
Type 1	3 (4)	1 (7)	2 (3)	0.85 ^5^
Type 2	54 (72)	11 (69)	43 (72)	
Type 3	9 (12)	2 (13)	7 (12)	
Type 4	9 (12)	1 (7)	8 (13)	
**Size**	n = 66	n = 16	n = 50	
Total tumor (mm)	37 ± 11	39 ± 12	37 ± 12	0.64 ^4^
Solid part (mm)	14 ± 6	14 ± 6	14 ± 6	0.96 ^4^
**Hydrocephalus**	n = 66	n = 16	n = 50	0.29 ^5^
Yes	18 (27)	6 (38)	12 (24)	
No	48 (73)	10 (62)	38 (76)	
**Location of other hemangioblastomas, if any ^1^**	n = 81	n = 18	n = 63	
Intracranial location	5 (6)	3 (17)	2 (3)	**0.04 ^6^**
Spinal location	9 (11)	1 (6)	8 (12)	0.12 ^6^
Retinal location	6 (7)	4 (22)	2 (3)	0.07 ^6^
**Management**	n = 80	n = 17	n = 63	
**Preoperative Embolization**				0.60 ^6^
Yes	1 (1)	0 (0)	1 (2)	
No	79 (99)	17 (100)	62 (98)	
**Extent of resection**				
Gross total resection	75 (94)	13 (72)	62 (98)	**0.0006 ^6^**
Partial resection	5 (6)	4 (18)	1 (2)	
**Radiotherapy**				
Yes	1 (1)	0 (0)	1 (2)	0.60 ^6^
No	79 (99)	17 (100)	62 (98)	
**Histological Features**				
**Aspect of the tumor stroma ^2^**	n = 74	n = 14	n = 60	
Reticular	58 (78)	9 (64)	49 (82)	**0.04 ^5^**
Cellular	20 (27)	5 (38)	15 (25)	0.64 ^6^
**Lipid-laden stromal cells**	n = 75	n = 15	n = 60	0.91 ^5^
Absence	36 (48)	7 (47)	29 (48)	
Presence	39 (52)	8 (53)	31 (52)	
**Biological Parameters**				
**Hematocrit level** (%) mean ±SD	n = 73	n = 13	n = 60	
Preoperative	42 ± 5	42 ± 5	40 ± 5	0.94 ^4^
Postoperative	39 ± 4	38 ± 7	29 ± 5	0.12 ^4^
**Survival Data**				
**Progression-free survival** (months)	n = 81	n = 18	n = 63	
	132	56	Not reached	**<0.0001 ^7^**
	95%CI [108;154]	95%CI [1;240]		
**Distant progression**	n = 81	n = 18	n = 63	**0.002 ^6^**
Yes	11 (14)	7 (39)	4 (6)	
No	70 (86)	11 (61)	59 (94)	

Data are expressed as count (percentage), unless otherwise specified. ^1^ Some patients had several locations. ^2^ Sometimes, several tumor stroma subtypes were found in different areas of the same tumor. ^3^ Comparison of patient with and without local tumor progression. ^4^ Mann-Whitney U-test. ^5^ Two-tailed χ^2^ test. ^6^ Fischer exact test. ^7^ Log-rank test. Significant *p*-values are indicated in bold characters.

**Table 2 cancers-16-01261-t002:** Radiological features and management of patients with local and distant progression.

	Local Progression n = 18	Distant Progression n = 11
**Resche Classification at progression**	n = 15	n = 10
Type 1	0 (0)	0 (0)
Type 2	11 (73)	2 (20)
Type 3	4 (27)	8 (80)
Type 4	0 (0)	0 (0)
**Management at progression**	n = 17	n = 11
Surgery only	11 (64)	5 (45.5)
Surgery + radiotherapy	2 (12)	0
Radiotherapy only	0	1 (9)
Follow-up	4 (24)	5 (45.5)

**Table 3 cancers-16-01261-t003:** Comparison of the characteristics of patients with and without VHL disease.

	VHL n = 11	Not VHL n = 70	*p*-Value
**Demographic Data**			
**Age (years) mean ± SD**	36 ± 14	50 ± 15	**0.0045 ^2^**
**Radiological Features**			
**Location of other hemangioblastomas if any ^1^**	n = 11	n = 70	
Total	18	2	**<0.0001 ^3^**
Intracranial location	5 (45%)	0 (0%)	**<0.0001 ^3^**
Spinal location	8 (73%)	1 (1%)	**<0.0001 ^3^**
Retinal location	5 (45%)	1 (1%)	**<0.0001 ^3^**
**Management**			
**Extent of resection**	n = 10	n = 70	
Gross total resection	10 (91%)	65 (93%)	1 ^3^
Partial resection	0 (0%)	5 (7%)	
**Histological Features**			
**Aspect of the tumor stroma**	n = 7	n = 67	
Reticular	6 (86%)	52 (78%)	0.62 ^4^
Cellular	1 (14%)	19 (28%)	0.43 ^3^
**Survival Data**			
**Progression-free survival** (months)	n = 11	n = 70	
	132	120	0.237 ^5^
	95%CI [104;163]	95%CI [88;148]	
**Local Progression**	n = 11	n = 70	
Yes	8 (73%)	10 (14%)	**<0.0001 ^3^**
No	3 (27%)	60 (86%)	
**Distant Progression**	n = 11	n = 70	**<0.0001 ^3^**
Yes	8 (73%)	3 (4%)	
No	3 (27%)	67 (96%)	

^1^ Some patients had several locations. ^2^ Mann–Whitney U-test. ^3^ Fischer exact test. ^4^ Two-tailed χ^2^ test. ^5^ Log-rank test. Significant *p*-values are indicated in bold characters.

**Table 4 cancers-16-01261-t004:** Factors associated with distant progression.

	Distant Progression n = 11	No distant Progression n = 70	*p*-Value
**Demographic Data**			
**Age (years) mean ± SD**	37 ± 17	50 ± 15	**0.01 ^2^**
**VHL disease**	8 (67)	3 (4)	**<0.0001 ^3^**
**Radiological Features**			
**Location of other hemangioblastomas if any ^1^**	n = 11	n = 70	
Intracranial location	4 (36)	1 (2)	**<0.001 ^3^**
Spinal location	6 (54)	3 (4)	**<0.0001 ^3^**
Retinal location	5 (46)	1 (2)	**<0.0001 ^3^**
**Aspect of the tumor stroma**	n = 8	n = 66	
Reticular type	7 (87)	51 (77)	0.51 ^4^
Cellular Type	1 (13)	19 (29)	0.51 ^3^
**Presence of lipid-laden stromal cells**	5 (63)	31 (47)	1 ^4^
**Local progression**	n = 11	n = 70	**0.002 ^4^**
Yes	7 (64)	11 (15)	
No	4 (36)	59 (85)	
**Follow-up (months and range)**	90	36	**0.025 ^2^**
	[24–240]	[1–204]	

^1^ Some patients had several locations. ^2^ Mann–Whitney U-test. ^3^ Fischer exact test. ^4^ Two-tailed χ^2^ test. Significant *p*-values are indicated in bold characters.

## Data Availability

The data presented in this study are available upon reasonable request from the corresponding author (Thiébaud Picart).

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
