# Peer review of "Predictors of Progression in a Series of 81 Adult Patients Surgically Managed for an Intracranial Hemangioblastoma: Implications for the Postoperative Follow-Up"

_cancers, 2024, doi:10.3390/cancers16071261_

Round 1

Reviewer 1 Report

Comments and Suggestions for Authors

Comments to authors-2024-03-07

Submission ID; cancers-2911633 

Thank you very much to give me an opportunity to review this manuscript.

Authors described prediction factors of local recurrence of hemangioblastoma (HGB). The paper was well-written. And I agree to authors’ opinion and current data, demonstrating that partial resection, young age, multiple lesions, and VHL were risk factors for local recurrence.

And as authors described, long-term follow-up for at least 10 years should be necessary even after gross total resection of the tumor, especially in case of VHL. I also experienced cumbersome case of HGB with VHL. The patient underwent three times surgical operation every 10 years! And gamma knife was not effective.

I would like to ask authors to additional description about radio-resistance of HGB and molecular target therapy of HIF-2 alpha inhibitor, Belzutifan.

Authors described best result of radiotherapy for HGB was that “without history of surgical resection”. So is postoperative adjuvant radiotherapy not recommended? I would like to ask authors to previous literature regarding postoperative adjuvant chemo-radiotherapy including gamma knife and molecular targeted therapy such as HIF-2a antagonist.

Minor point

In table 2, the number of “distant progression” should be “n=11” instead of “n=12”.

I think this should be corrected.

Author Response

We thank reviewer 1 for her/his positive feedback and for the relevant comments.

  1. Surgery remains the first therapeutic option for the management of hemangioblastomas, especially in sporadic cases but also in patients with VHL, at least for symptomatic locations. Regarding the results associated to postoperative radiotherapy, few results are available. However, according to the survival analysis led by Yin et al, patients receiving surgery for the management of a cranial or spinal hemangioblastoma had a better overall survival compared with patients receiving both surgery and radiotherapy (Yin et al). Consequently, postoperative adjuvant radiotherapy should not be recommended systematically and only be discussed in patients with very partial resection. Gammaknife is mainly proposed in patients with a VHL disease with multiple locations in the central nervous sytem. In this indication, the rate of local failure is low (4%) (Liebenow et al). A systematic review, also led in patients with VHL disease, consistently concluded that gamma-knife could represent an interesting option in patients with limited surgical alternative, leading to a tumor control in more than 90% of cases at 5-years. However, the long-term outcomes and underlying factors associated with tumor progression remain to be investigated (Qiu et al).

HIF-2α inhibitor were approved by the FDA in 2021, notably for patients with VHL disease with hemangioblastomas located in the central nervous system. These drugs are associated with a 30% response rate in hemangioblastomas and a reduction in their sizes from 20% to 50% over the course of treatment. Anemia is the most common adverse effect. Although long-term results are not known, these treatments represent a promising option (Neth et al, Zamarud et al, Zhang et al).

These elements were added in the discussion (§ 4.5 Recommendations regarding patient management)

  1. Minor point: we acknowledge that there was a typo. We apologize for this mistake that was corrected.

Reviewer 2 Report

Comments and Suggestions for Authors

The authors present a retrospective review of 81 patients with hemangioblastoma and evalauted perioperative factors which controibuted to rates of recurrence in this population. Overall this is a high quality manuscript that is well written and the methodology is sound. It will additionally be of interest to the readership and will allow other clinicians better care for these patients in the future. There are no glaring omissions nor any significant issues that I have with this report. My only point to make is that while patients with VHL were considered to "recurrent" lesions at local or distant sites from the first surgical region, the authors must acknowledge that these may be new sporadic lesions in these patients rather than true recurrences. This is a minor point, but it should be made. I would be happy with this manuscript if this point is acknowledged. Well done. 

Author Response

We thank Reviewer 2 for this positive feedback and for giving us the opportunity to discuss the problematic of the concept of recurrence in this particular disease.

We totally agree that it was not possible to discriminate between true local recurrence or new sporadic lesion in patients with VHL. This element was added in the discussion (§4.2 Genetic predisposition of hemangioblastoma progression).